# Identify specific gene pairs for subarachnoid hemorrhage based on wavelet analysis and genetic algorithm

**Pengcheng Zhao, Shaonian Xu, Zhenshan Huang, Pengcheng Deng, Yongming Zhang** * 

Department of Neurosurgery, Anhui No. 2 Provincal People's Hospital, Hefei, Anhui, China

* zymhf2966@163.com

## Abstract

Subarachnoid hemorrhage (SAH) is a fatal stroke caused by bleeding in the brain. SAH can be caused by a ruptured aneurysm or head injury. One-third of patients will survive and recover. One-third will survive with disability; one-third will die. The focus of treatment is to stop bleeding, restore normal blood flow, and prevent vasospasm. Treatment for SAH varies, depending on the bleeding's underlying cause and the extent of damage to the brain. Treatment may include lifesaving measures, symptom relief, repair of the bleeding vessel, and complication prevention. However, the useful diagnostic biomarkers of SAH are still limited due to the instability of gene marker expression. To overcome this limitation, we developed a new protocol pairing genes and screened significant gene pairs based on the feature selection algorithm. A classifier was constructed with the selected gene pairs and achieved a high performance.

## Introduction

Intracranial aneurysm is one of the most clinically dangerous cerebrovascular diseases. The ruptured aneurysm forms subarachnoid hemorrhage (aneurysmal subarachnoid hemorrhage, aSAH) or combined with intracranial hematoma, which is disabling and has a high fatality rate. The 30-day mortality rate of patients with aSAH was 45%, and 30% of survivors had moderate to severe disability. The exact pathogenesis of the development of IA and subsequent aSAH is not yet clear, but processes such as hemodynamic stress, matrix degeneration, and inflammation seem to play a role. About 10% of aSAH patients have one or more first-degree relatives with aSAH, and unaffected first-degree relatives have an increased risk of aneurysm and aSAH. IA usually has no symptoms before it ruptures, so it must be screened. CT angiography (CTA) is the current standard screening method for developing atherosclerosis and subsequent rupture in high-risk populations. Still, screening has disadvantages in terms of cost and negative consequences. Therefore, we need tools better to detect high-risk groups of aneurysm development or rupture. In a recent study, Xu et al compared the mRNA expression in whole blood of SAH patients with and without vasospasm. The results demonstrate that mRNA expression level signatures can be applied to distinguish SAH patients from normal controls [1]. Another study also evaluated the relationship between aSAHpatient outcomes and genetic variants and DNA methylation [2]. The author found that DNA methylation of

**Funding:** Key Research and Development Projects in Anhui Province (No. 201904a07020108).

**Competing interests:** The authors have declared that no competing interests exist.

hepcidin geneplayscritical roles in patients following aSAH. Both studies imply the potential application of gene signatures as diagnostic and prognostic biomarkers for aSAH patients.

The traditional feature gene screening method classifies samples based on the expression level of a single gene. Although the training data's fitting effect is excellent, it involves cross-platform issues, and the inconsistency of experimental procedures and analysis objects results in a large verification set. It is difficult to achieve the expected accuracy. Although the expression of a single gene fluctuates significantly, the relationship between a pair of related genes is relatively stable; that is, no matter how the expression of the two genes fluctuates, the difference between the two genes is always the same. This feature can be used as a more stable diagnostic feature. Therefore, this study uses Fourier transform wavelet analysis and genetic algorithm to screen out significant gene association pairs, establish a deep learning model to predict subarachnoid hemorrhage, and realize the early clinical diagnosis of subarachnoid bleeding caused by arterial rupture diagnosis and prediction.

This research uses a wavelet analysis algorithm to realize feature selection and combines genetic algorithms to construct a diagnosis model. Our model has high diagnostic efficiency. It shows that genetic relationships can make an early prediction of cerebral hemorrhage patients more robust and accurate. In addition, since the features we use are binary relationships of gene relationship pairs, there is no need to consider different platforms' batch effects. Considering that the relative quantitative relationship between two genes is stable on any platform, our features and models can be applied to data on other platforms without the need for additional standardization and batch effect correction. This study's contributions include the new features of gene pairs and the diagnostic model built via these new features. The combination of gene pairs gives insight to a better understanding of the pathogenic mechanism of SAH.

## Methods

### 1. Expression profile processing

We downloaded the expression profile data GSE73378 of human subarachnoid hemorrhage from the GEO database. The data contained 226 samples, including 112 aSAH patients and 114 normal controls. The sequencing platform is Illumina HumanHT-12 V4.0 expression beadchip. To make all genes comparable at the same distribution level, we calibrate all expression values through a z-test [3]. The calibration process includes calculating the expression mean μ and standard deviation (SD) in the control group, and then the expression values of all samples are corrected.

### 2. Differentially expressed genes

We used normal samples as the control group and aSAH patients as the case group. The gene significance was calculated by the limmaRpackage [4]. Limma is a package for the analysis of gene expression data arising from microarray or RNA-seq technologies. It uses linear models to assess differential expression in the context of multifactor designed experiments. Finally, genes with a significant P value of less than 0.01 and logFC>1 or logFC<-1 were used as substantial differences expressed genes. The distribution of differentially expressed genes in background genes is visualized using volcano maps.

### 3. Cluster analysis

To further observe the difference between cerebral hemorrhage-related genes in disease state and normal state, we use differentially expressed genes to perform hierarchical clustering of all cerebral hemorrhage samples and normal control samples. The clustering process is

implemented by the R heatmap package [5]. Hierarchical clustering is an unsupervised clustering algorithm that can illustrate the diversity between groups based on gene expression. The similarity matrix uses the Pearson correlation coefficient algorithm [6] and finally is visualized in the form of a heat map.

## 4. Principal component analysis

To identify a set of genes significantly associated with subarachnoid hemorrhage from among the many differentially expressed genes, we first performed PCA dimensionality reduction treatment on the genes [7]. PCA is different from hierarchical clustering since it first extracts several major components through orthogonal decomposition and then presents the sample distribution. We can identify the number of principal components with the most considerable explanation variance through principal component analysis, representing the distribution of association sets within genes. Simultaneously, we use the mclust algorithm [8] to cluster the genes, determine the number of clusters, and evaluate the model's clustering effect through the BIC criterion [9]. Combining principal component analysis and mclust, we finally identified the associated groups within genes.

## 5. Functional analysis using the specific genes

We use principal component analysis and mclust clustering algorithm to cluster genes based on gene expression level correlation and identify gene sets with expression correlation. These expression-related genes often have functional consistency, so we use these specific genes. The function enrichment analysis was performed on the set. The enrichment method was performed by DAVID software [10], and the KEGG functional pathway and Gene Ontology (GO) term with a significant P value less than 0.05 were taken as the essential functions. These functional annotation tools are integrated to illustrate the abnormal functions regulated by specific genes.

## 6. Pearson correlation analysis

We used Pearson correlation to screen out robust gene relationship pairs with vasospasm sensitivity as diagnostic markers to calculate any two differentially expressed genes. If the correlation coefficient is more significant than 0.5 and the significance p-value is less than 0.05, this is considered. There is a positive correlation between the two genes; if the correlation coefficient is less than -0.5 and the significance p-value is less than 0.05, then the two genes are considered to be negatively correlated. To visually observe the correlation distribution of differentially expressed genes, we use the pheatmap r package to draw a correlation matrix's heat map. The positive correlation and the negative correlation are represented by red and green color patches, respectively.

## 7. Coexpression network construction

In the complex regulatory network related to genes, some genes act as sources and regulate multiple downstream genes simultaneously through physical or genetic interactions. On the contrary, certspecifices act as targets, fulfilling their functions precisely under the control of upstream genes. Some of these genespresentsingle regulation, while some present compound regulation from multiple sources. To quantify the interaction regulation relationship between genes and determine the hub genes that can participate in the regulation of various genes at the same time or receive multi-party compound regulation, we use Pearson correlation to calculate the co-expression correlation between any two genes. Gene pairs with a coefficient

greater than 0.5 are regarded as positive correlations, and gene pairs with correlation coefficients less than -0.5 are considered to be negative correlations. Based on the co-expression correlation between genes, we constructed a gene co-expression similarity network. Nodes represent genes, and edges represent interactions between genes, with red up-regulation and green down-regulation. By analyzing the network topology, the hub gene in the network can be identified. These genes have a high degree of mediation between the network and possible primary disease targets or diagnostic markers.

## 8. Identify significant gene association pairs based on wavelet analysis

First of all, the genes are divided into two groups based on high expression and low expression to ensure that the expression of each group of genes is relatively consistent, that is, the differential expression direction of adjacent genes is the same, so it is easy to find rules among them. Sort the genes according to the clustering results. In the same cluster, the genes are sorted from low to high in the co-expression network. This ensures that the importance of two adjacent genes in the regulatory network is relatively close. It may have similar functional positioning. To reduce background noise and make the expression data more stable, we calculated each gene's average expression in the control combination case group, performed a difference operation on all the classified genes, and finally used a moving average to reduce system fluctuations. We used wavelet transform [11] to analyze the gene expression data after the noise reduction process and identify the specific expression patterns and significant gene association pairs in each gene cluster (prominent peaks indicate substantial differences in the expression patterns of two adjacent genes). Wavelet transform is a feature selection algorithm that can select the most significant gene pairs.

## 9. Feature selection of genes using genetic algorithm

The genetic algorithm [12] simulates the biological evolution process. Through the process of parental chromosome recombination, the offspring with poor adaptability are eliminated, and the offspring with strong adaptability are amplified to optimize the most suitable combination of genetic information. We randomly combine all identified gene association pairs to form a "feature chain" and initialize the feature chain's length to 50% of the total number of features. Then extract a pair of feature chains for reorganization so that the offspring feature chains also contain feature information from the parent and use highly adaptable offspring to calculate the fitness of the progeny (fitness is the prediction accuracy of the feature chain in this study). The feature chain eliminates the offspring's low-adaptive characteristic chain, and this process continues in a loop until the maximum evolutionary algebra is reached or the model converges to find the best characteristic chain. Then, gradually reduce the initial feature chain's length, repeat the above process, and finally obtain the best feature-related gene pair combination.

## 10. Deep learning model

We use genetic algorithms to perform evolutionary screening of feature-related gene pairs and obtain feature combinations that are significantly related to subarachnoid hemorrhage. We used the difference between these genes' expression values as feature values to train and predict with the neural network deep learning model [13]. The neural network's initialization parameters are the activation function: sigmoid, learning rate 0.01, 5 units in the input layer, ten units in the hidden layer, and 1 unit in the output layer. We randomly sort the analysis data, take 50% as the training set, and the remaining 50% as the test set. The training process uses the gridsearch [14] algorithm to optimize the parameters. The optimized parameters include the

activation function, the number of hidden layers, and the learning rate. Finally, the ROC [15] curve is used to evaluate model classification and prediction performance.

## Results

### 1. Differentially expressed genes extraction

We tested the difference of all genes in the two sets of samples, and all gene expression values were corrected to a normal distribution based on the mean and variance of the control group. Finally, a total of 1453 genes, 855 up-regulated genes, and 598 down-regulated genes were identified. The distribution of differentially expressed genes is shown in Fig 1.

Among all genes, the proportion of up-regulated genes was 6.27%, and the ratio of down-regulated genes was 4.13%, which was close to the 5% interval of two-tailed normal distribution.

### 2. Cluster analysis

Based on the expression values of differentially expressed genes in the two sets of samples, we perform hierarchical cluster analysis on the samples to verify whether differentially expressed

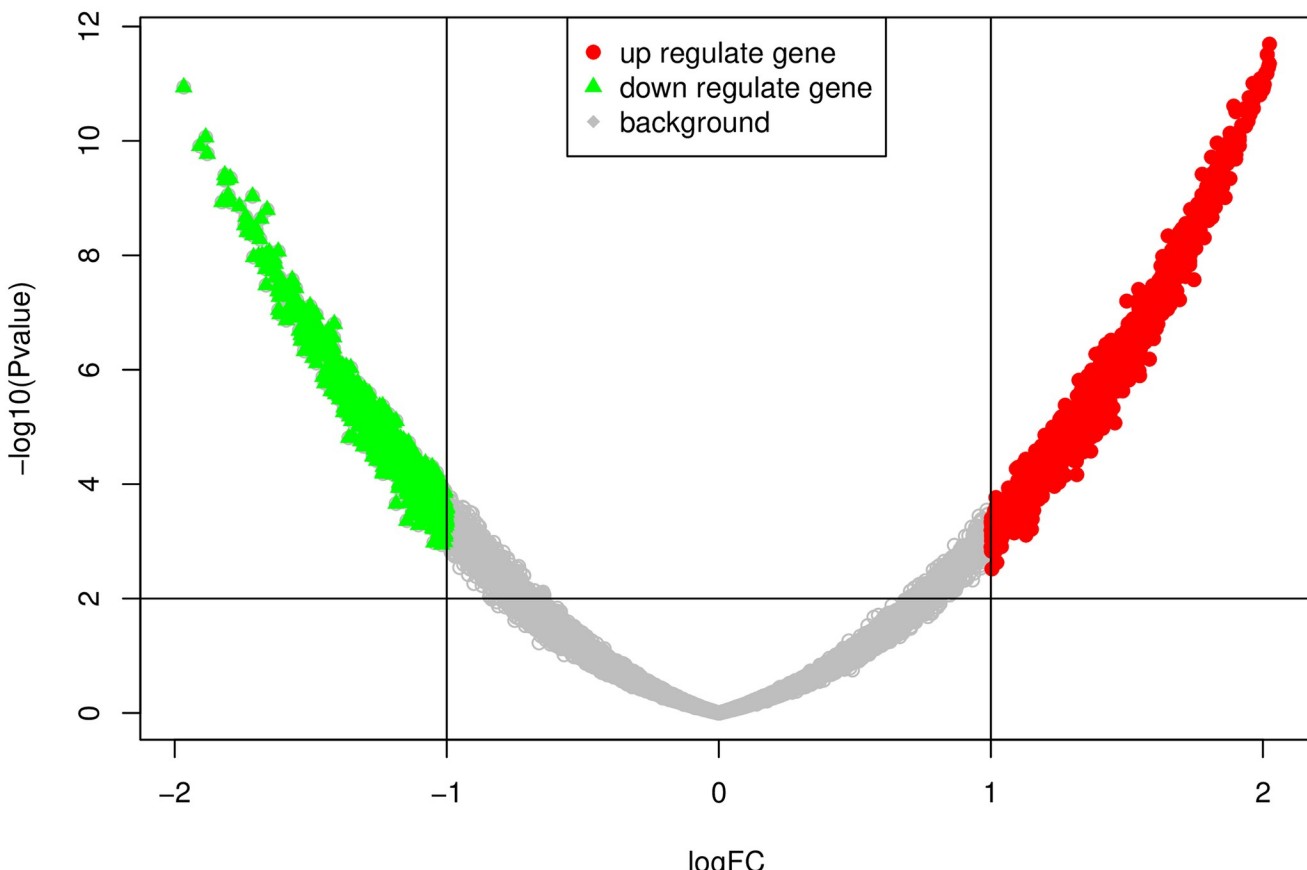

**Fig 1. Volcano graph.** The horizontal axis is logfc, and the vertical axis is the p-value of negative logarithmic transformation.

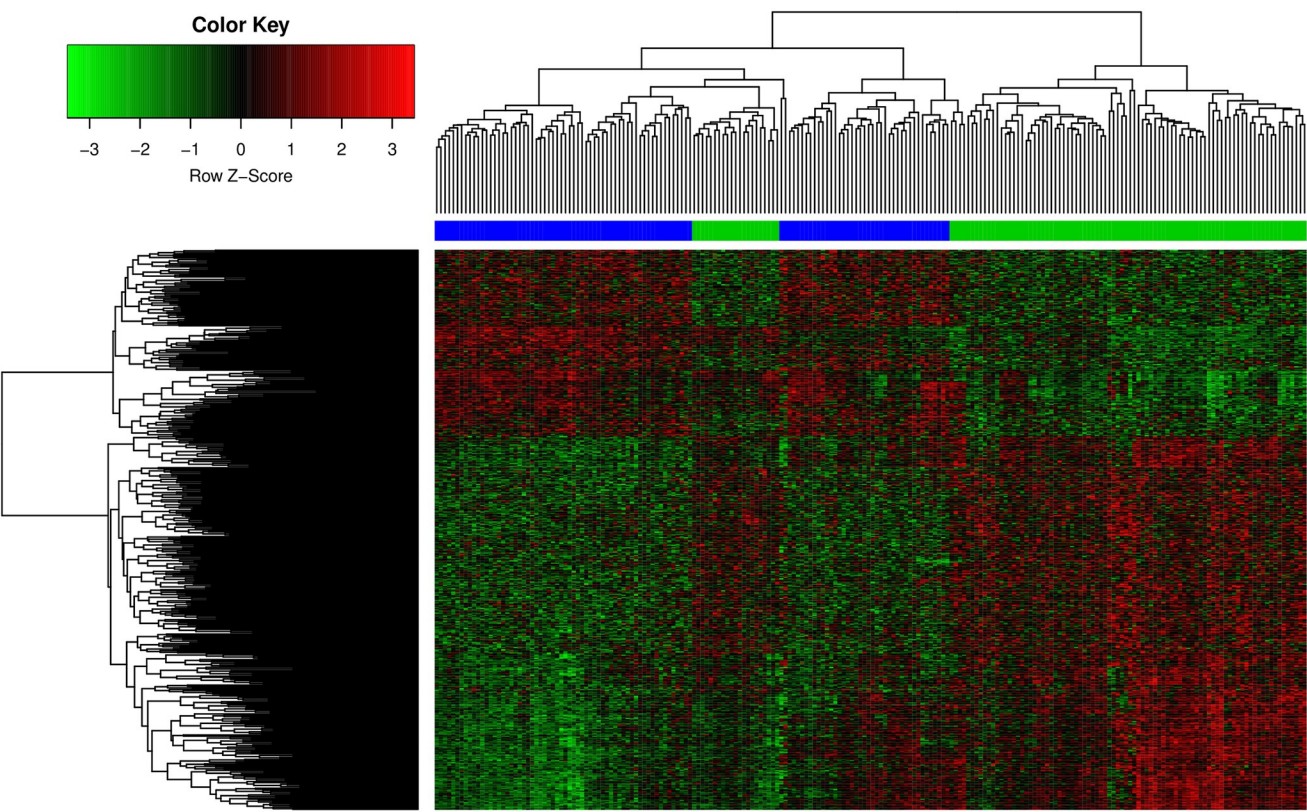

**Fig 2. Cluster heatmap.** Disease and normal sample are drawn with red and green, respectively.

genes have significantly different expression patterns in different groups of samples. The heat map of hierarchical clustering is shown in Fig 2.

From the results of the heat map, it can be observed that 1. Based on the differentially expressed genes, the two sets of samples can be accurately distinguished, but some control samples are mixed into the cerebral hemorrhage sample group; 2. The differentially expressed genes are significantly different between the samples of different groups. In the expression pattern, some genes are expressed as low expression in the case group (upper right quadrant) but become high expression in the control group (lower right quadrant). Conversely, some highly expressed genes in the case group (lower right quadrant) became lowly expressed in the control group (lower left quadrant). This result suggests that a more detailed gene subset can be further separated within the differentially expressed genes. The genes in the subset show strong co-expression correlation and suggest the inherent functional consistency of these genes.

## 3. Principal component analysis

To further analyze the entire differentially expressed gene set in more detail, we first use principal component analysis to reduce the gene set's dimensionality. The results of the principal component analysis are shown in Fig 3.

In Figure, the inflection point represents the inflection point's position, corresponding to 7 principal components, which means that when starting from the sevenprincipal components and increasing the principal components, the rate of decrease of eigenvalue is attenuated. The seven principal components are more reasonable.

**Parallel Analysis Scree Plots**

**Fig 3. Principle component analysis.** The blue and red curves represent the real and simulated data.

On the other hand, we also use the mclust algorithm for verification. We extract the first three principal components in PCA to cluster genes. The result is shown in Fig 4.

It can be seen that genes can be effectively divided into seven different subsets under the action of the first three principal components. At the same time, it was found that pc1 is the best forgene clustering, while pc2 and pc3 are weaker for gene clustering.

CombiningPCA and mclust, we finally divided the differentially expressed genes into seven different subsets. See the supplementary material geneclass.txt for the results.

## 4. Functional enrichment

The genes in each subset show obvious co-expression correlation, and the co-expression of genes also reflects the functional consistency between genes to a certain extent. To explain each subset's specific biological function, we use each subset to perform available enrichment analysis on the genes. The enrichment content includes the KEGG pathway and the go term. Enrichment analysis is performed on each gene subset by DAVID software. We take the most significant top3 features of each subgroup for research, as shown in the Fig 5.

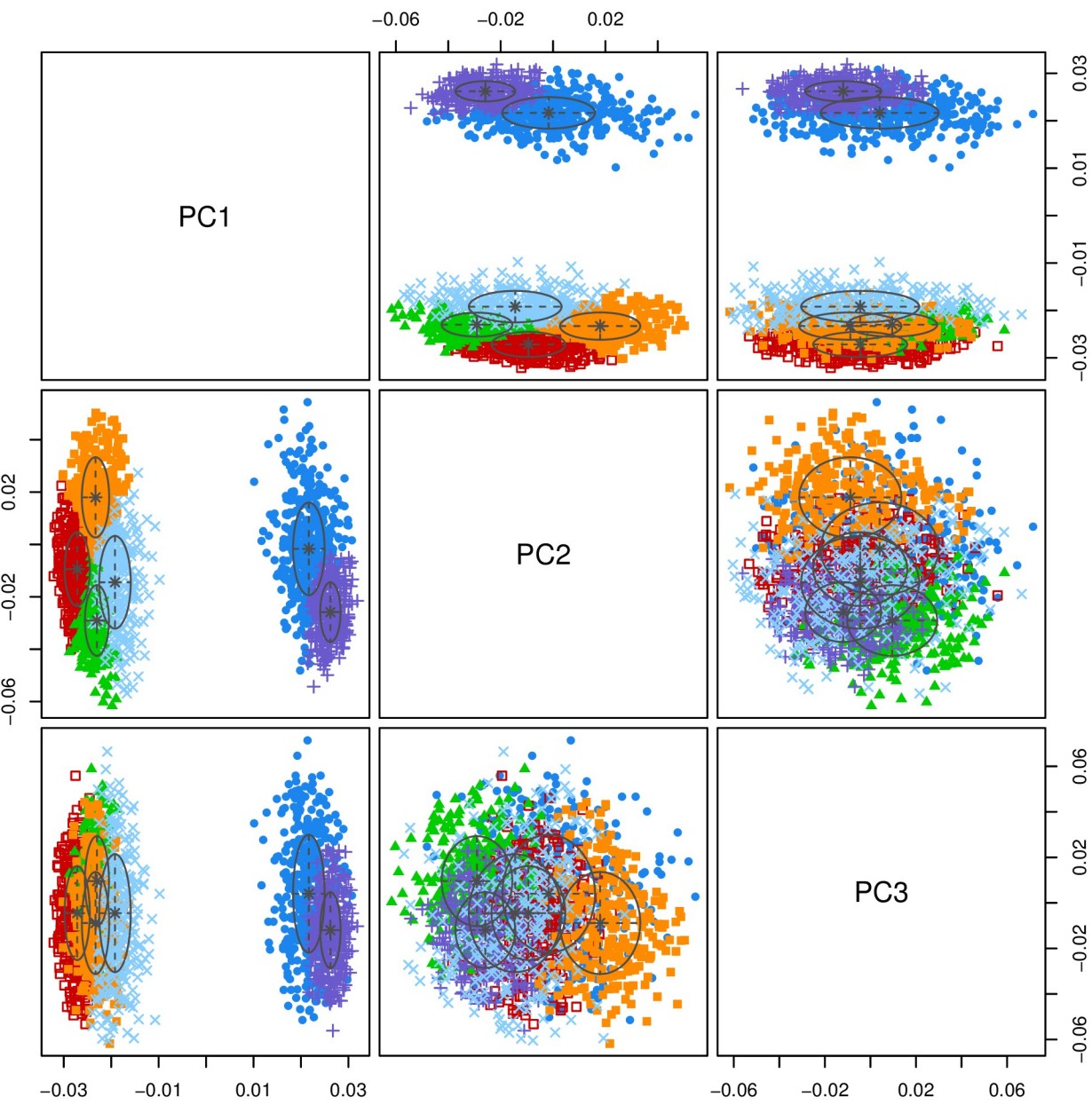

**Fig 4. Mclust graph.** The x-axis and y-axis represent the top three components. Different groups are marked with different colors.

In Fig 5, the horizontal axis is the number of genes enriched to each function, and the vertical axis is the function term. We use seven colors to mark seven different gene sets. According to the most significant top3 function of each subset selected (gene set6 only enriched to two functions), it can be observed that each gene subset has a specific function annotation. For example, the gene set1 focuses on the cell generation stage, including cytoskeleton and microtubule formation. Geneset2 focuses on the insulin pathway, autophagy, and mRNA process. Gene set3 concentrates on the interaction between cells, including platelet activity, cell adhesion, lipopolysaccharide response, etc. Gene set4 focuses on transcription regulation, including positive regulation, negative regulation, ubiquitination, and so on. Gene set5 focuses on hormones, chemokines, and other body fluid regulation. Gene set6 focuses on protein stability

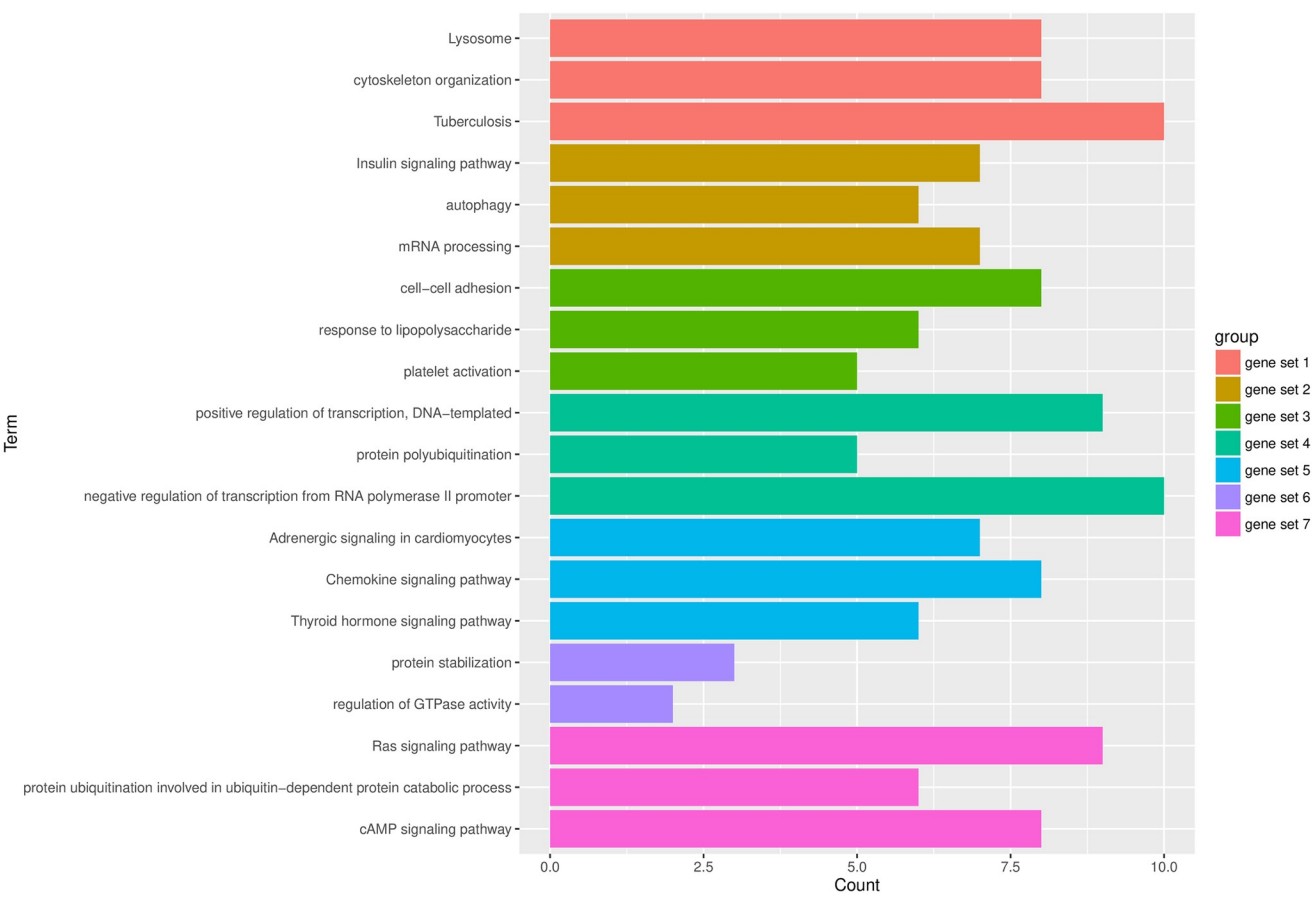

**Fig 5. The most significant functions in seven gene sets.** The x-axis represents the gene count and the y-axis represents the functions regulated by each gene class.

and GTP energy metabolism. Gene set7 focuses on the RAS pathway and the camp pathway, both common diseases and cancer-related pathways.

## 5. Coexpression network construction

We pair all differentially expressed genes in any pair and calculate the correlation of a pair of genes in all samples. After random pairing, there are 400,000 gene pairs with a correlation coefficient greater than 0.5. Most of these relationship pairs are gene pairs that have no significant biological correlation. Therefore, to screen out the genuinely possible correlation pairs, we strictly enforce the Pearson correlation coefficient. Restriction, that is, relationship pairs with a correlation greater than 0.8, are filtered out, as shown in the Fig 6.

We use the correlation between gene pairs as edges and genes as nodes to construct a co-expression network. The network contains 832 gene nodes and 8681 edges. We set the colors of the edges to purple and green based on the positive and negative correlations and set the nodes to red and green based on the high and low gene expression, as shown in Fig 7.

According to the distribution of gene nodes in the network, it can be observed that some genes have a co-expression correlation with multiple genes at the same time, indicating that these genes may act as regulators and simultaneously regulate the expression of numerous downstream genes, or these genes themselves are target genes. From the compound regulation

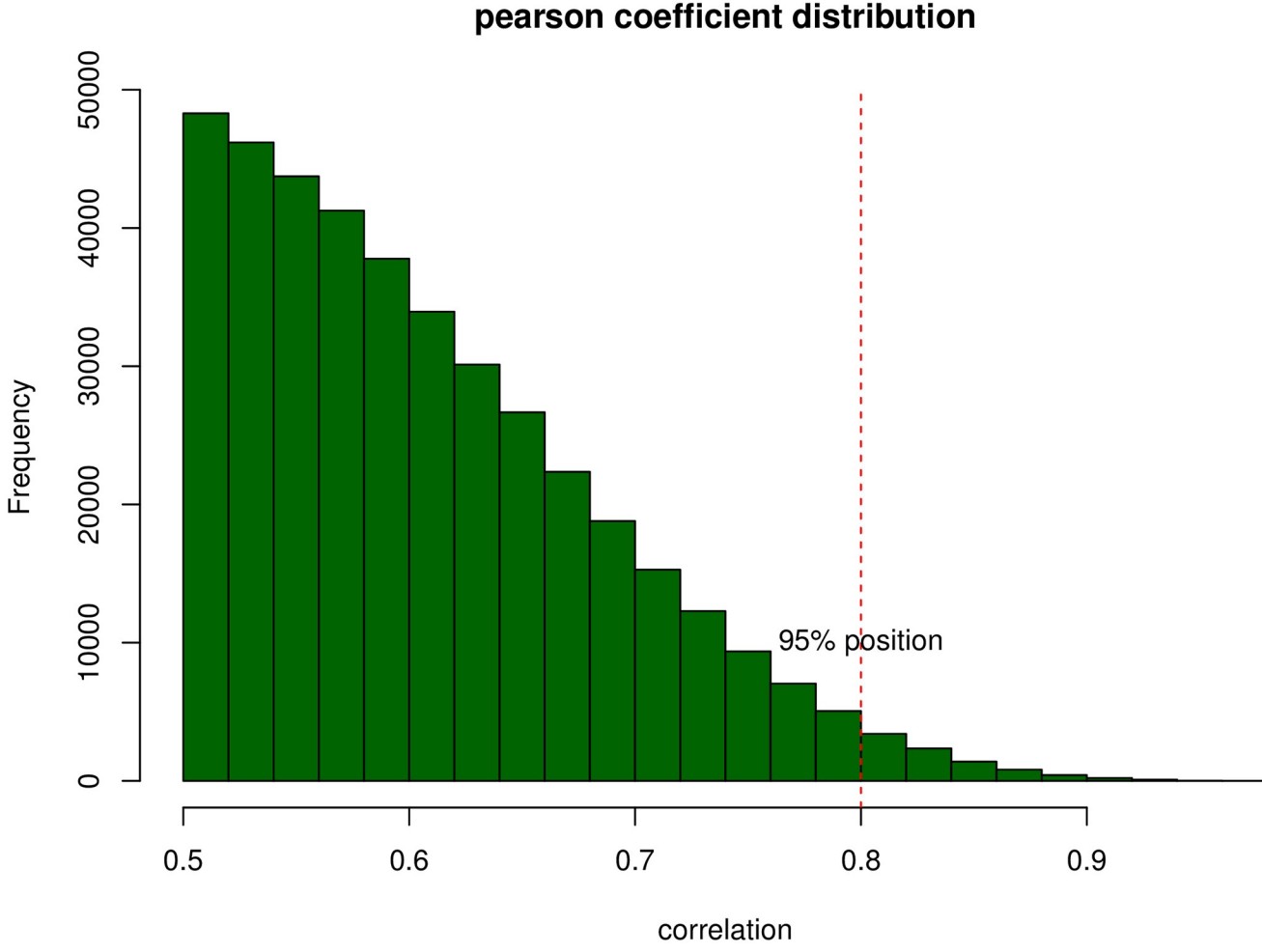

**Fig 6. Pearson coefficient distribution.** The x-axis represents the correlation efficiency and the y-axis represents the frequency.

of multiple genes, these hub genes with higher nodes are likely to have an essential subarachnoid hemorrhage correlation.

Fig 8 shows the distribution of node degree in the network. The horizontal axis is the node degree transformed by the negative logarithm, and the vertical axis is the density distribution. It can be seen that there are two apparent peaks in the distribution of nodes. The first one is degree 1, the gene whose degree is 0 after negative logarithm transformation, and the other has a degree of 5, which is 2.32 after negative logarithm transformation.

## 6. Wavelet analysis

We sort the genes according to the subsets they belong to and the genes' node degree within each subgroup. Two adjacent genes are related at the expression level and fairness in biological importance. After difference and moving average processing, wavelet analysis is used to process the expression signal to identify each gene subset's expression pattern and specific gene association pairs. We treat up-regulated and down-regulated genes separately, as shown in Fig 9.

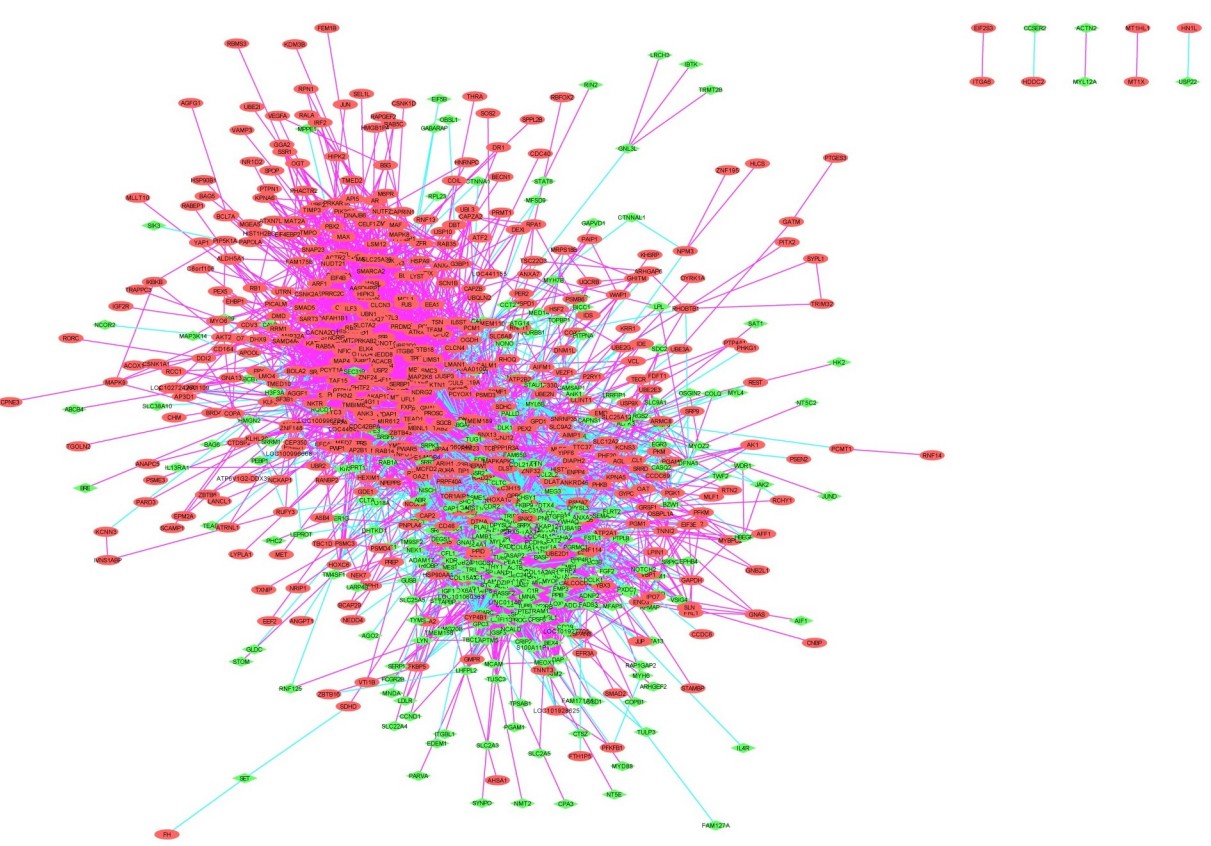

**Fig 7. Co-expression network.** The red and green nodes are up and down-regulated genes.

We use the red vertical line to tell the genes are divided into seven sets according to gene subsets. The left picture is the original signal expression value, and the right picture is the signal value after wavelet transformation. Firstly, it can be observed that the signals of up-regulated genes and down-regulated genes in different sample groups and different subsets have different patterns. First, in the case group, the wavelet analysis of the wavelet significance threshold marked by the green dashed line; in the case group, the threshold is close to 0.1, while the threshold value in the control group is close to 0.001, so whether up-regulated or down-regulated genes, there are more apparent fluctuations than in the control group. For example, the up-regulated genes of gene set1 show more obvious volatility, while the down-regulated genes of gene set2 show more apparent fluctuations. To identify the significant gene association pairs in each subset, we take the gene pairs corresponding to the significant peaks exceeding the four-group analysis threshold as features.

## 7. Feature extraction based on genetic algorithm

Based on the significantly related gene pairs extracted by wavelet analysis, we use the difference between the expression values of the two genes in the gene pair as the feature value, and perform feature selection through genetic algorithm, and finally screen out five features with fitness greater than 0.5, as shown in Fig 10.

We listed the fitness scores of the top gene pairs in Table 1. Feature stands for gene pair, and fitness stands for fitness. Therefore, we use these five related gene pairs as features to construct a model.

## node degree Density

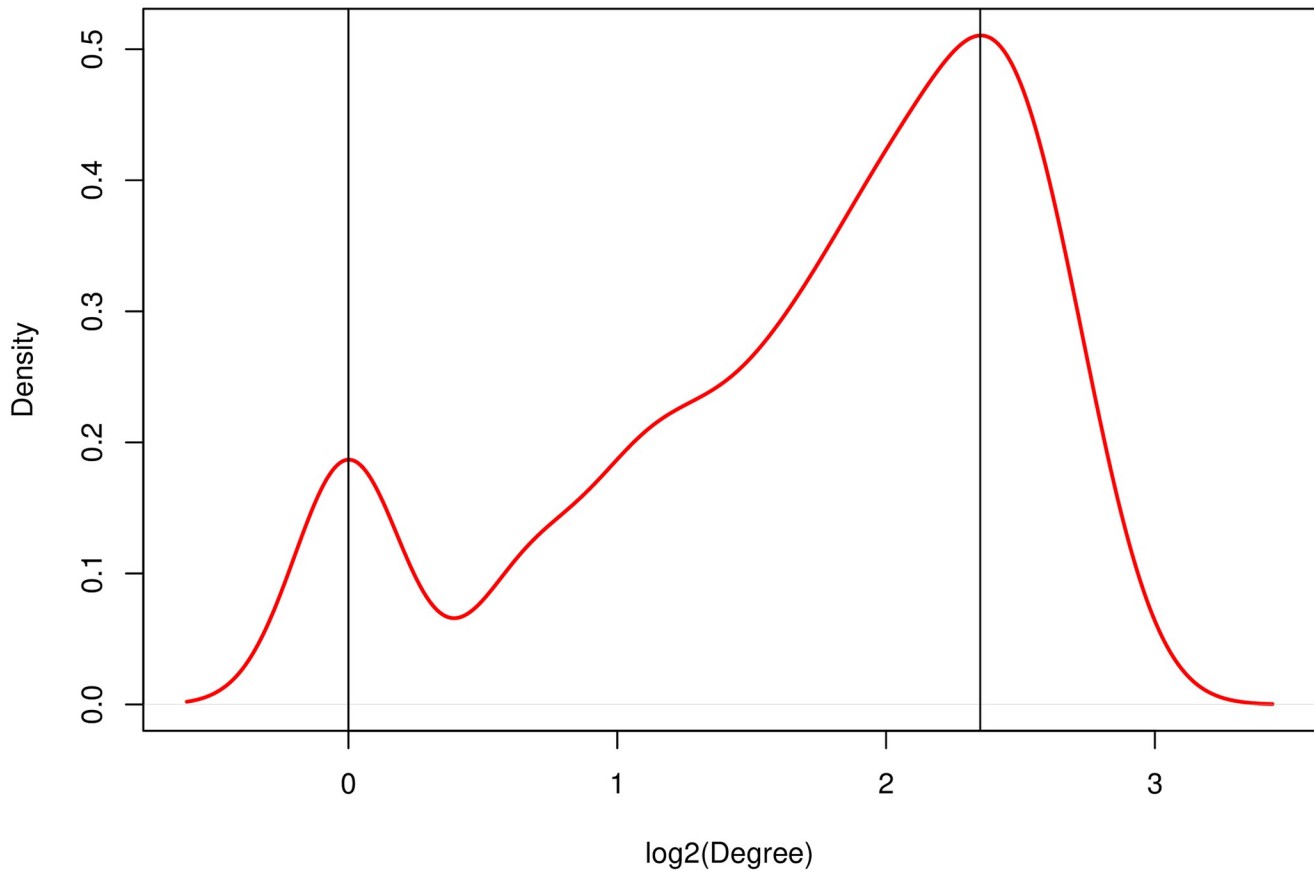

**Fig 8. Degree distribution of nodes.** The x-axis represents the log-transformed degree and the y-axis represents the density.

## 8. Deep learning model construction

We use a genetic algorithm to optimize and screen out significant feature-related genes to build the model. All model parameters are initialized. After grid search parameter optimization, the model parameters are changed to activation function relu, learning rate of 0.1, and two hidden layers. It consists of 10 and 5 units. After the samples are randomly rearranged, 50% of them are used to train the model, and the remaining 50% are tested. The results are represented by the roc curve, as shown in the Fig 11.

The horizontal axis is specificity. The vertical axis is sensitivity. The red line is the accuracy of the training set, and the green line is the accuracy of the test set. After feature selection and parameter optimization, the training set's accuracy and test set reach 93% and 87%. On the one hand, the model accuracy reached a high level. On the other hand, the training set results and the test set are relatively close, indicating that the model has not overfitted. Therefore, the five associated gene pairs that we have identified, using the difference in expression of two genes between the gene pairs, can accurately identify patients with cerebral hemorrhage and normal controls.

On the one hand, it meets the diagnostic requirements, and on the other hand, it overcomes the shortcomings of single gene labeling are large fluctuations and low reliability. At the same

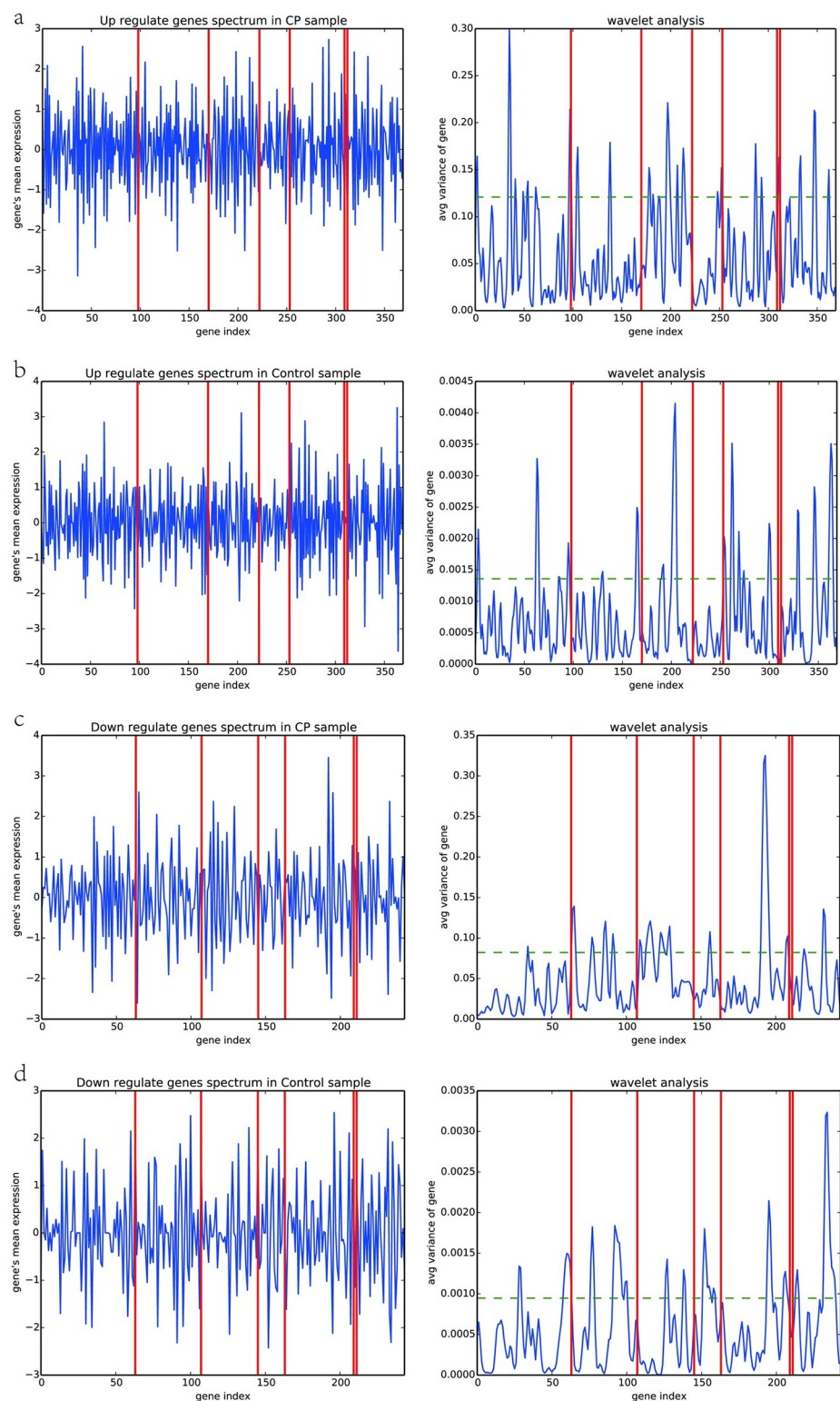

**Fig 9. (a) up genes in the case group; (b) up genes in the control group; (c) down the gene in case group; (d) down the gene in the control group.**

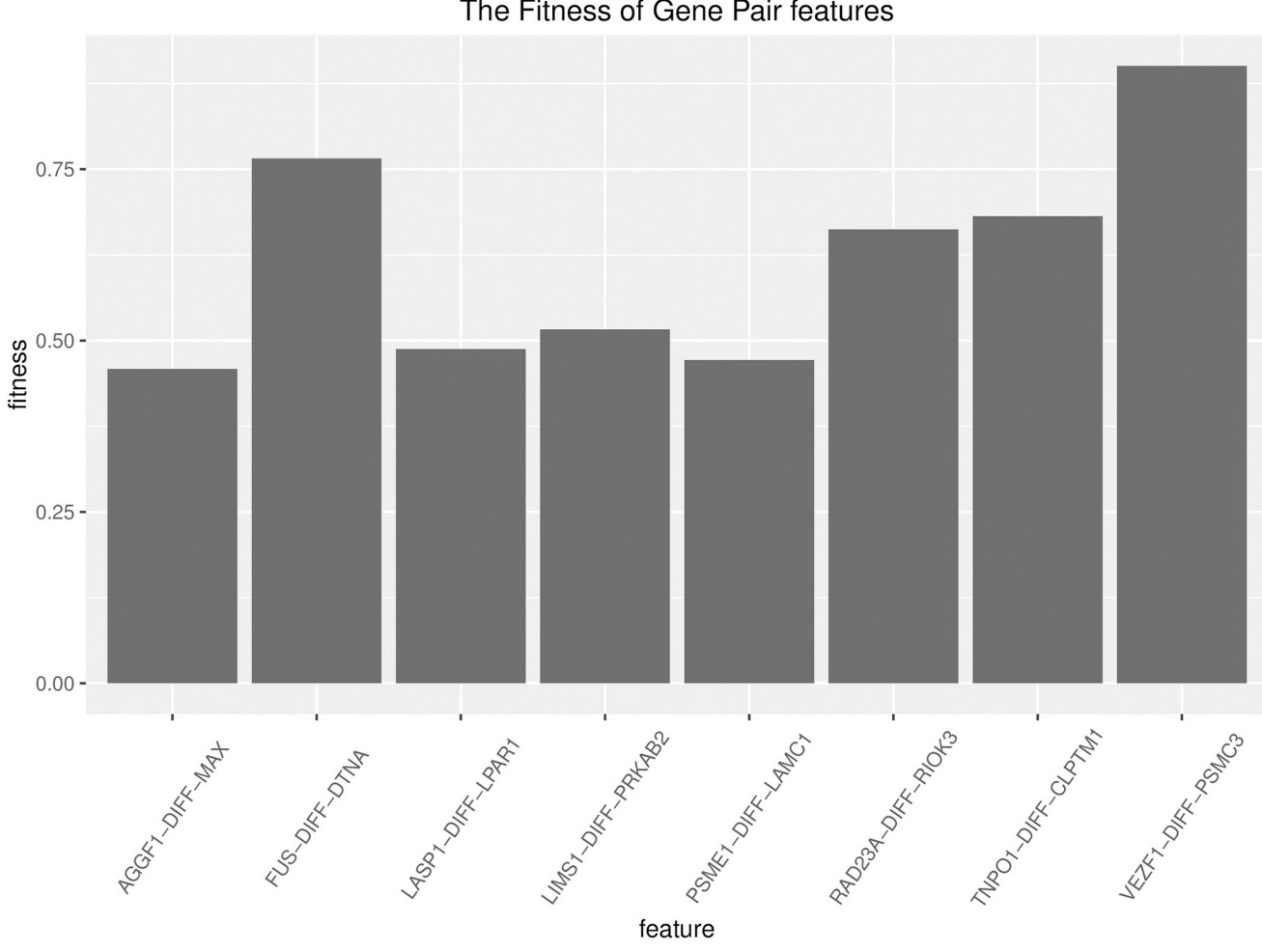

**Fig 10. Feature fitness.** The x-axis represents the features and the y-axis represents the fitness.

time, these related gene pairs indicate that there is a biologically possible dysfunction related to cerebral hemorrhage. Under normal circumstances, the difference between the two genes is not apparent. Still, this difference is due to one gene or two genes malfunctioned simultaneously in the disease course. They were amplified, which also suggested a new pathogenic mechanism for patients with cerebral hemorrhage.

## Discussion

ASAH is an acute, high-risk, and highly insidious disease. Its incidence is increasing year by year, and its disability and mortality are high. If it is not detected and controlled in time, it will

**Table 1. significant gene pairs.**

| feature | Fitness |
|---|---|
| VEZF1-PSMC3 | 0.9007576 |
| FUS-DTNA | 0.7656250 |
| RAD23A-RIOK3 | 0.6621429 |
| TNPO1-CLPTM1 | 0.6814655 |
| LIMS1-PRKAB2 | 0.5162693 |

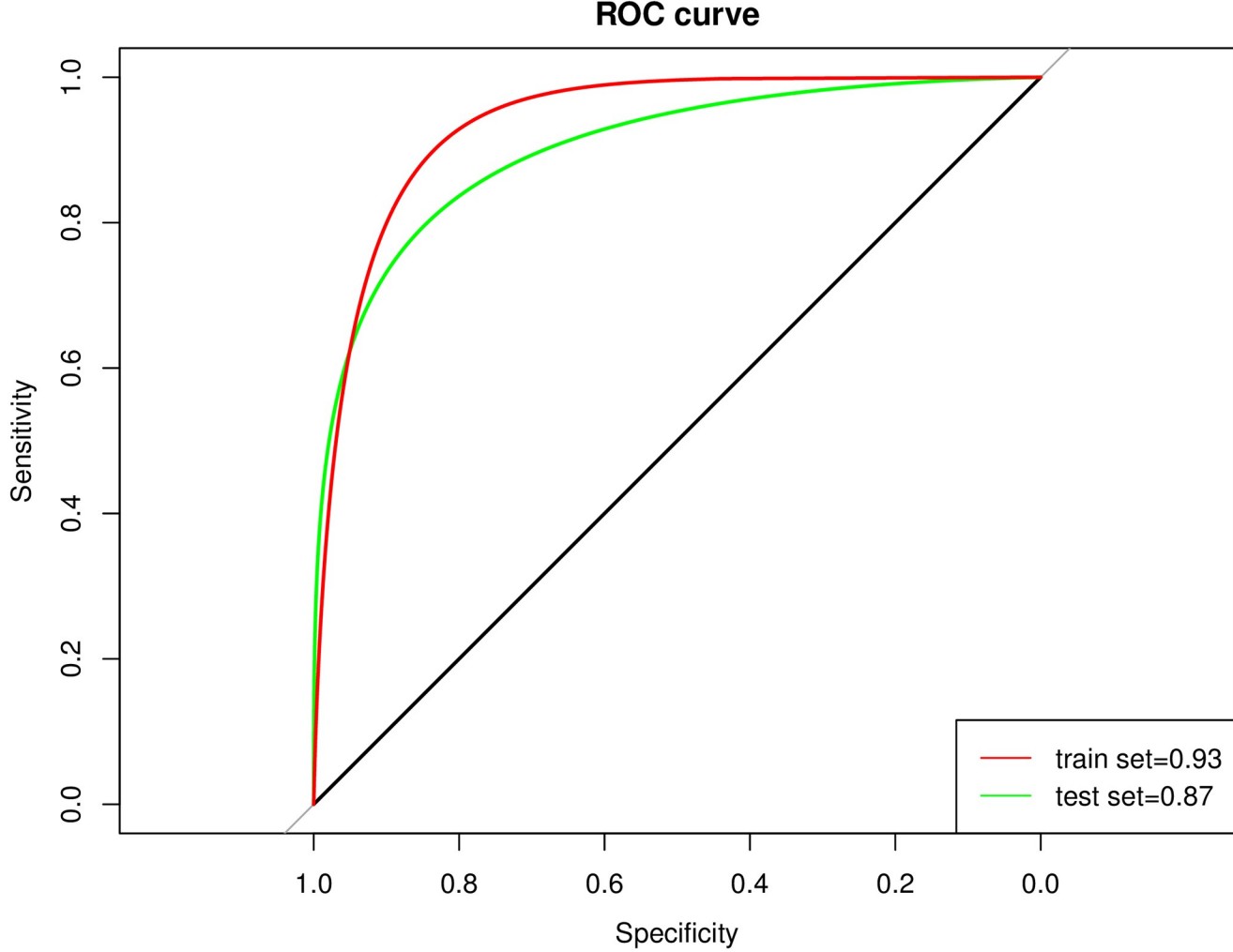

**Fig 11. Roc curve.** The x-axis and y-axis represent the specificity and sensitivity, respectively. The red and green curves correspond to the train and test set.

cause severe brain tissue damage. Therefore, it is crucial to determine the markers in IA's occurrence and development and find intracranial aneurysms in a timely, effective, and accurate manner. In this study, by obtaining data from GEO, we determined the DEG between IA and ordinary people, combined with PCA and mclust, we finally divided the differentially expressed genes into seven different subsets and performed functions on each gene subset enrichment analysis, it can be observed that each gene subset has a specific functional annotation. These results suggest that different available modules may cause the formation and development of IA.

Then we pair all the differentially expressed genes in any pair and calculate the correlation of a pair of genes in all samples. Simultaneously, wavelet analysis and genetic algorithms were used to extract significant related gene pairs and perform feature selection. A total of five corresponding gene pairs were screened out, namely VEZF1-PSMC3, FUS-DTNA, RAD23A-RIOK3, TNPO1-CLPTM1, LIMS1-PRKAB2, VEZF1- PSMC3. We analyzed these genes in detail.

VEZF1 is a zinc-finger transcription factor that encodes during the vascular development of mouse embryos [16]. It is specifically expressed in endothelial cells. When VEZF1 is

inactivated in mice, mouse embryos' lethality increases, which seriously affects the vascular system. The differentiation and proliferation of endothelial cells [17]. Studies have shown that VEZF1 can regulate endothelin 1 (EDN1), matrix metalloproteinase 2 (MMP2), several genes related to angiogenesis [18], and Vezf1 also plays an essential role in regulating the expression of anti-angiogenic factor Cited2 [19]. Angiogenesis,blood vessels' formation, is necessary for normal development, but angiogenesis also leads to various diseases, including cancer and cardiovascular diseases. Studies have proved the role of Vezf1 in regulating the heart's compensatory growth and the contractile function of cardiomyocytes, which may be related to human heart disease [20]. The development of new blood vessels through the angiogenesis process is critical indeveloping and growing aneurysms. Therefore, Vezf1 may be related to the growth of aneurysms. PSMC3 encodes 26S regulatory subunit 6A, also known as the 26S proteasome AAA-ATPase subunit (Rpt5) of the 19S proteasome complex, which is responsible for recognizing, unfolding, and transporting substrates to the 20S protease cavity of the proteasome [21]. This protein family has various cellular functions, including cell cycle regulation, gene expression, vesicle-mediated transport, peroxisome assembly, and proteasome functions [22]. Studies have pointed out that PSMC3 is a critical apoptotic gene, which is significantly abundant in metabolism-related events (such as the metabolism of mRNA), related to caspase inhibitors, and is highlighted in the co-expression network [23].

The genetic mutation of FUS is a DNA/RNA binding protein that can cause frontotemporal degeneration (FTLD) and amyotrophic lateral sclerosis (ALS). FUS involves multiple aspects of RNA metabolism, including RNA splicing, transport, and translation. The factor shuttles between the nucleus and the cytoplasm. It can control more than 5500 identified RNA targets in the human and mouse brains. It has been widely studied in frontotemporal degeneration (FTLD) and amyotrophic lateral sclerosis (ALS). DTNA is a cytoplasmic protein related to cell signaling. The DTNA gene consists of 23 coding exons and is regulated by extensive splicing. As a component of the postsynaptic device on the synaptic membrane, DTNA is related to the anchoring of acetylcholine receptors. Also, it has been noted that DTNA, as a putative causative gene involved in left ventricular non-compact cardiomyopathy, can effectively regulate myocardial hypertrophy and microtubule structure.

RAD23A is an evolutionarily conserved protein, which is very important for nucleotide excision repair. It interacts with the 26S proteasome through the amino-terminal ubiquitin-like domain (UBL[R23]) [24]. HR23A and HR23B are human orthologs of the yeast Rad23 (HR23) protein. Studies have shown that HR23A critically regulates the expression of Twist1 protein [25] and Chk1 protein and plays a vital role in cell functions (such as cell cycle progression, immunity, and stress response). Human HR23A protein has been shown to help control the stability and transcriptional activity of p53 [26]. Studies on the mechanism of spinocerebellar ataxia type 3 neurodegeneration have demonstrated that exogenous RAD23 increases the toxicity of pathogenic ataxin-3. This is consistent with the rise in disease protein levels [27]. This shows that RAD23A plays a vital role in the regulation of various diseases. RIOK3 (RIO) kinase is a conserved atypical serine/threonine-protein kinase family. Evidence shows that RIO kinase plays an essential role in ribosomal biosynthesis in mammalian cells [28] and cancer cell proliferation, apoptosis, migration, and invasion. Down-regulation of RIOK3 significantly reduced the AKT/mTOR signaling pathway activity and induced apoptosis of glioma cells. Overexpression of RIOK3 has opposite effects on the proliferation, migration, invasion of glioma cells, and the AKT/mTOR signal transduction pathway [29].

Shipment protein 1, TNPO1, encodes the β subunit of the nuclear adhesin receptor complex. The specific interaction of nuclear localization signal (NLSS) can regulate the formation of membrane-less organelles and the subcellular localization of numerous proteins [30]. The study of Janiszewska provides evidence that the entry of full-length CD44 into the nucleus is

dependent onTNPO1 [31]. CD44 plays an essential role in developing and developing athero-sclerosis by mediating inflammatory cell recruitment and vascular cell activation. A study showed that [32], TNPO1 may play an indispensable role in developing atherosclerosis by regulating the transport of CD44.

Cleft lip and palate transmembrane protein 1 (Clptm1), a multichannel transmembrane protein that seems to be commonly expressed, including in the brain [33]. Clptm1 restricts the forward transport of GABAR and modulates the steady-state plasticity of inhibition [34]. We found that Clptm1 interacts with multiple recombinant GABAAR subunits and unusually regulates phase and phase by limiting receptor surface expression. Tonic inhibitory transmission and simulates homeostasis inhibitory synaptic scaling. The risk of lung cancer in non-smokers is related to gene mutations in the CLPTM1L1 region [35].

LIMS1 (also known as PINCH1) was initially thought to be a kind of self-epitope protein homologous to "senescent cell antigen," which is widely expressed in mammalian cells [36] and can be linked to integrin-linked kinase (ILK) and the ternary complex composed of Parvin is connected to the actin cytoskeleton and many different signaling pathways, and plays a regulatory role in gene transcription or cell adhesion [37]. Hypoxia induces the expression of the LIMS1 gene on the cell surface [38], and it plays an essential role in promoting the growth of diseased cells in the microenvironment of hypoxia-glucose deprivation [39]. LIMS1 has been reported to be overexpressed in different types of tumors and is even an independent prognostic factor [40]. PrKAB2 (AMPKB2) is a gene encoding β-2 regulatory subunit located at 1q21.2. Northern blot analysis showed that AMP-activated protein kinase-β2 is expressed in the form of 7.5kb mRNA in various human tissues [41]. The expression level of PRKAB2 in patients with coronary heart disease and type 2 diabetes has always been a research hotspot [42]. AMPK is essential in maintaining the integrity of Drosophila neurons in the nervous system and stress, survival, and longevity. Loss of AMPKβ may increase the risk of mental disorders and sleep disorders associated with human 1q21.1 loss [43].

Simultaneously, to check the model's accuracy, we use genetic algorithms to optimize and screen out significant feature-related genes to build the model. The accuracy of the training set and the test set reaches 93% and 87%. On the one hand, the model accuracy reaches a high level. The training set results and the test set are relatively close, indicating that the model has not been overfitted. Therefore, the five related gene pairs that we have identified, using the expression difference between the two genes between the gene pairs, can accurately identify patients with cerebral hemorrhage and normal controls. On the other hand, it meets the diagnostic requirements and also overcomes shortcomings such as large fluctuations in single gene expression and low reliability. At the same time, these related gene pairs indicate that there is a biologically possible dysfunction related to cerebral hemorrhage. Under normal circumstances, the difference between the two genes is not apparent. Still, this difference is due to one gene or two genes malfunctioned simultaneously in the disease course. They were amplified, which also suggested a new pathogenic mechanism in patients with cerebral hemorrhage.

This study also has some limitations. First, we highlighted the interaction between genes and adopted this relationship as new feature units. However, the interaction could be more complex, and thus,a feature unit is likely composed of more than two genes. This advanced relationship is not considered in our study yet. Second, the relationship between genes is restricted to the expressional absolute ranking. There are multiple gene interactions such as physical binding, transcriptional regulation, inherit interaction,etc. Taking these kinds of interactions into account may increase the biological interpretation and prediction accuracy. All these limitations are beyond our current study's scope but will be involved in our future work.

## Conclusion

This paper uses wavelet analysis and genetic algorithm to screen out significant gene association pairs and establishes a deep learning model to predict subarachnoid hemorrhage. On the one hand, these related genes can be used as new diagnostic markers or drug targets; on the other hand, they can also heuristically predict potential gene regulation relationships. This regulatory relationship is significantly different between patients with cerebral hemorrhage and normal controls, further explaining the possible pathogenic subarachnoid bleeding mechanism.

## Supporting information

**S1 Table. Gene class by PCA and mclust.**
(TXT)

## Author Contributions

**Conceptualization:** Pengcheng Zhao, Shaonian Xu, Zhenshan Huang, Yongming Zhang.

**Data curation:** Pengcheng Zhao, Shaonian Xu, Zhenshan Huang, Pengcheng Deng, Yongming Zhang.

**Formal analysis:** Pengcheng Zhao, Pengcheng Deng, Yongming Zhang.

**Funding acquisition:** Pengcheng Zhao, Yongming Zhang.

**Investigation:** Pengcheng Zhao, Shaonian Xu, Yongming Zhang.

**Methodology:** Pengcheng Zhao, Shaonian Xu, Zhenshan Huang, Pengcheng Deng, Yongming Zhang.

**Project administration:** Pengcheng Zhao, Shaonian Xu, Zhenshan Huang, Yongming Zhang.

**Resources:** Pengcheng Zhao, Yongming Zhang.

**Software:** Pengcheng Zhao, Shaonian Xu, Pengcheng Deng, Yongming Zhang.

**Supervision:** Pengcheng Zhao, Shaonian Xu, Yongming Zhang.

**Validation:** Pengcheng Zhao, Shaonian Xu, Zhenshan Huang, Yongming Zhang.

**Visualization:** Pengcheng Zhao, Shaonian Xu, Zhenshan Huang, Pengcheng Deng, Yongming Zhang.

**Writing – original draft:** Pengcheng Zhao, Shaonian Xu, Zhenshan Huang, Pengcheng Deng, Yongming Zhang.

**Writing – review & editing:** Pengcheng Zhao, Shaonian Xu, Zhenshan Huang, Pengcheng Deng, Yongming Zhang.

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
