## [Editor Report · Decision Letter 0]

18 Jan 2021

PONE-D-20-40638

Identify specific gene pairs for subarachnoid hemorrhage based on wavelet analysis and genetic algorithm

PLOS ONE

Dear Dr. Zhang,

Thank you for submitting your manuscript to PLOS ONE. After careful consideration, we feel that it has merit but does not fully meet PLOS ONE’s publication criteria as it currently stands. Therefore, we invite you to submit a revised version of the manuscript that addresses the points raised during the review process.

Before review process, some comments should be addressed for manuscript revision.

Therefore, I invite the authors to resubmit the revised manuscript for further reviews.

1. The authors should address the format of PLOS ONE to revise the whole paper.

2. The research questions should be defined.

3. The contributions of this study should be highlight in the first section.

4. A literature review section should be given and discussed.

5. The authors should recognize the section of method. They only listed some ideas in the second section. However, they should present the method and how the method can solve the research questions. The quality of presentation should be improved. 

6. The authors should recognize the section of results. They only listed some results in the third section. However, they should discuss the results and why the method can solve the research questions. The quality of presentation should be improved. 

7. The authors should present the limitation of this study.

8. Future work should be discuss and given in the last section.

We look forward to receiving your revised manuscript.

Kind regards,

Chi-Hua Chen, Ph.D.

Academic Editor

PLOS ONE

Additional Editor Comments:

Before review process, some comments should be addressed for manuscript revision.

Therefore, I invite the authors to resubmit the revised manuscript for further reviews.

1. The authors should address the format of PLOS ONE to revise the whole paper.

2. The research questions should be defined.

3. The contributions of this study should be highlight in the first section.

4. A literature review section should be given and discussed.

5. The authors should recognize the section of method. They only listed some ideas in the second section. However, they should present the method and how the method can solve the research questions. The quality of presentation should be improved. 

6. The authors should recognize the section of results. They only listed some results in the third section. However, they should discuss the results and why the method can solve the research questions. The quality of presentation should be improved. 

7. The authors should present the limitation of this study.

8. Future work should be discuss and given in the last section.

Journal Requirements:

2. Please note that PLOS ONE does not copy edit accepted manuscripts (https://journals.plos.org/plosone/s/criteria-for-publication#loc-5). To that effect, please ensure that your submission is free of typos and grammatical errors, including the title. Please also refer to our submission guidelines (https://journals.plos.org/plosone/s/submission-guidelines#loc-human-subjects-research), and note that outmoded terms and potentially stigmatizing labels, such as "surviving among the disabled" should be changed to more current, acceptable terminology.

*PLOS ONE has specific criteria for papers that describe new methods or software for applications. Specifically, these reports must meet the criteria of utility, validation, and availability, which are described in detail at http://journals.plos.org/plosone/s/submission-guidelines#loc-methods-software-databases-and-tools. Please ensure that you detail in your methods section in what way the method presented in your manuscript represents a proven advantage over existing alternatives.

3. We note you have included a table to which you do not refer in the text of your manuscript. Please ensure that you refer to Table 1 in your text; if accepted, production will need this reference to link the reader to the Table.

5. Thank you for submitting the above manuscript to PLOS ONE. During our internal evaluation of the manuscript, we found significant text overlap between your submission and the following previously published works, on some of which you may be an author.

https://onlinelibrary.wiley.com/doi/abs/10.1002/jcp.28442

https://www.eurekaselect.com/172331/article

https://pubs.acs.org/doi/10.1021/acs.biochem.6b01134

Please revise the manuscript to rephrase the duplicated text, cite your sources, and provide details as to how the current manuscript advances on previous work. Please note that further consideration is dependent on the submission of a manuscript that addresses these concerns about the overlap in text with published work.

---

## [Author Response · Author response to Decision Letter 0]

23 Mar 2021

1. The authors should address the format of PLOS ONE to revise the whole paper.

Reply to editor: Thanks for your suggestions. We have revised the manuscript according

to the MANUSCRIPT BODY FORMATTING GUIDELINES. The font and text size

have been modified accordingly.

2. The research questions should be defined.

Reply to editor: We highlighted this study's aim and the issues we want to address in the

abstract section. The primary subjective is to find out new biomarkers for subarachnoid

hemorrhage and build a diagnostic predictor.

3. The contributions of this study should be highlight in the first section.

Reply to editor: Thanks for your suggestion. We have included the contributions of this

study to the abstract section. The significant contribution is that we developed a new

protocol pairing genes and screened significant gene pairs based on the feature selection

algorithm. A classifier was constructed with the selected gene pairs and achieved a high

performance.

4. A literature review section should be given and discussed.

Reply to editor: We included the literature review in the discussion part. Multiple studies

are discussed and verified our signature genes. The conclusion of this study is well

concordant to the earlier studies.

5. The authors should recognize the section of the method. They only listed some ideas in

the second section. However, they should present the method and how the method can

solve the research questions. The quality of the presentation should be improved.

Reply to editor: The method section has been reorganized. We added the description

regarding how each method can solve the research questions.

6. The authors should recognize the section of the results. They only listed some results in

the third section. However, they should discuss the results and why the method can solve

the research questions. The quality of the presentation should be improved.

Reply to editor: The result section is also revised. We added descriptions about why the

result contributes to this study.

7. The authors should present the limitation of this study.

Reply to editor: The limitation of this study has been included in the discussion.

8. Future work should be discussed and given in the last section.

Reply to editor: We stated some drawbacks of this study, and even though they are out of

the scope of our current study, they will be included in our future work.

Finally,Thank you very much for your comments on the language of the article, and we have modified it again. However, considering the professionalism and rigor of the language, if you think there are still some problems with the language, I would like to ask if your journal has a channel for polishing it. If so, please tell me the link, thank you. If you have any other questions, please contact me and thank you again.

---

## [Decision Letter · Decision Letter 1]

14 Apr 2021

PONE-D-20-40638R1

Identify specific gene pairs for subarachnoid hemorrhage based on wavelet analysis and genetic algorithm

PLOS ONE

Dear Dr. Zhang,

Thank you for submitting your manuscript to PLOS ONE. After careful consideration, we feel that it has merit but does not fully meet PLOS ONE’s publication criteria as it currently stands. Therefore, we invite you to submit a revised version of the manuscript that addresses the points raised during the review process.

We look forward to receiving your revised manuscript.

Kind regards,

Chi-Hua Chen, Ph.D.

Academic Editor

PLOS ONE

Journal Requirements:

Reviewers' comments:

Reviewer's Responses to Questions

**Comments to the Author**

1. If the authors have adequately addressed your comments raised in a previous round of review and you feel that this manuscript is now acceptable for publication, you may indicate that here to bypass the “Comments to the Author” section, enter your conflict of interest statement in the “Confidential to Editor” section, and submit your "Accept" recommendation.

Reviewer #1: (No Response)

2. Is the manuscript technically sound, and do the data support the conclusions?

Reviewer #1: (No Response)

3. Has the statistical analysis been performed appropriately and rigorously? 

Reviewer #1: (No Response)

4. Have the authors made all data underlying the findings in their manuscript fully available?

Reviewer #1: (No Response)

5. Is the manuscript presented in an intelligible fashion and written in standard English?

Reviewer #1: (No Response)

6. Review Comments to the Author

Reviewer #1: Interesting.

Please consider discussing your findings within the context of these two recent publications:

Xu, H., Stamova, B., Ander, B.P. et al. mRNA Expression Profiles from Whole Blood Associated with Vasospasm in Patients with Subarachnoid Hemorrhage. Neurocrit Care 33, 82–89 (2020).

Heinsberg, L.W., Arockiaraj, A.I., Crago, E.A. et al. Genetic Variability and Trajectories of DNA Methylation May Support a Role for HAMP in Patient Outcomes After Aneurysmal Subarachnoid Hemorrhage. Neurocrit Care 32, 550–563 (2020).

7. PLOS authors have the option to publish the peer review history of their article (what does this mean?). If published, this will include your full peer review and any attached files.

Reviewer #1: No

---

## [Author Response · Author response to Decision Letter 1]

27 May 2021

1. Please review your reference list to ensure that it is complete and correct.

Reply to editor: Thanks for your suggestions. We have checked the reference list and found no retracted papers.

2. Please consider discussing your findings within the context of these two recent publications:

…

Reply to editor: Thanks for your suggestions. We read the above two articles carefully, Xu et al compared the mRNA expression in whole blood of SAH patients with and without vasospasm. The results demonstrate that mRNA expression level signatures can be applied to distinguish SAH patients from normal controls. Another study also evaluated the relationship between aSAH patient outcomes and genetic variants and DNA methylation. The author found that DNA methylation of hepcidin geneplayscritical roles in patients following aSAH. Both studies imply the potential application of gene signatures as diagnostic and prognostic biomarkers for aSAH patients. We have added the above in the introduction(Manuscript Page2) and have added references from both articles to our list of references(Manuscript Page13).

3. Please include a separate legend for each figure in your manuscript.

Reply to editor: Thanks for your suggestions.I have added all the legends at the end of this article.

---

## [Decision Letter · Decision Letter 2]

31 May 2021

Identify specific gene pairs for subarachnoid hemorrhage based on wavelet analysis and genetic algorithm

PONE-D-20-40638R2

Dear Dr. Zhang,

We’re pleased to inform you that your manuscript has been judged scientifically suitable for publication and will be formally accepted for publication once it meets all outstanding technical requirements.

Kind regards,

Chi-Hua Chen, Ph.D.

Academic Editor

PLOS ONE

Additional Editor Comments (optional):

Reviewers' comments:

Reviewer's Responses to Questions

**Comments to the Author**

1. If the authors have adequately addressed your comments raised in a previous round of review and you feel that this manuscript is now acceptable for publication, you may indicate that here to bypass the “Comments to the Author” section, enter your conflict of interest statement in the “Confidential to Editor” section, and submit your "Accept" recommendation.

Reviewer #1: (No Response)

2. Is the manuscript technically sound, and do the data support the conclusions?

Reviewer #1: (No Response)

3. Has the statistical analysis been performed appropriately and rigorously? 

Reviewer #1: (No Response)

4. Have the authors made all data underlying the findings in their manuscript fully available?

Reviewer #1: (No Response)

5. Is the manuscript presented in an intelligible fashion and written in standard English?

Reviewer #1: (No Response)

6. Review Comments to the Author

Reviewer #1: The authors have adequately responded to prior comments/critiques. Thank you.

7. PLOS authors have the option to publish the peer review history of their article (what does this mean?). If published, this will include your full peer review and any attached files.

Reviewer #1: No

---

## [Editor Report · Acceptance letter]

7 Jun 2021

PONE-D-20-40638R2 

Identify specific gene pairs for subarachnoid hemorrhage based on wavelet analysis and genetic algorithm 

Dear Dr. Zhang:

I'm pleased to inform you that your manuscript has been deemed suitable for publication in PLOS ONE. Congratulations! Your manuscript is now with our production department. 

Kind regards, 

on behalf of

Professor Chi-Hua Chen 

Academic Editor

PLOS ONE